# On the Parameterization and Initialization of Diagonal State Space Models

**Albert Gu[†], Ankit Gupta[‡], Karan Goel[†], Christopher Ré[†]**
[†] Department of Computer Science, Stanford University
[‡] IBM Research
{albertgu,knrg}@stanford.edu, chrismre@cs.stanford.edu
ankitgupta.iitkanpur@gmail.com

## Abstract

State space models (SSM) have recently been shown to be very effective as a deep learning layer as a promising alternative to sequence models such as RNNs, CNNs, or Transformers. The first version to show this potential was the S4 model, which is particularly effective on tasks involving long-range dependencies by using a prescribed state matrix called the HiPPO matrix. While this has an interpretable mathematical mechanism for modeling long dependencies, it introduces a custom representation and algorithm that can be difficult to implement. On the other hand, a recent variant of S4 called DSS showed that restricting the state matrix to be fully diagonal can still preserve the performance of the original model when using a specific initialization based on approximating S4's matrix. This work seeks to systematically understand how to parameterize and initialize such diagonal state space models. While it follows from classical results that almost all SSMs have an equivalent diagonal form, we show that the initialization is critical for performance. We explain why DSS works mathematically, by showing that the diagonal restriction of S4's matrix surprisingly recovers the same kernel in the limit of infinite state dimension. We also systematically describe various design choices in parameterizing and computing diagonal SSMs, and perform a controlled empirical study ablating the effects of these choices. Our final model S4D is a simple diagonal version of S4 whose kernel computation requires just 2 lines of code and performs comparably to S4 in almost all settings, with state-of-the-art results for image, audio, and medical time-series domains, and averaging 85% on the Long Range Arena benchmark.

## 1 Introduction

A core class of models in modern deep learning are sequence models, which are parameterized mappings operating on arbitrary sequences of inputs. Recent approaches based on state space models (SSMs) have outperformed traditional deep sequence models such as recurrent neural networks (RNNs), convolutional neural networks (CNNs), and Transformers, in both computational efficiency and modeling ability. In particular, the S4 model displayed strong results on a range of sequence modeling tasks, especially on long sequences [9]. Its ability to model long-range dependencies arises from being defined with a particular state matrix called the "HiPPO matrix" [6], which allows S4 to be viewed as a convolutional model that decomposes an input onto an orthogonal system of smooth basis functions[10].

However, beyond its theoretical interpretation, actually computing S4 as a deep learning model requires a sophisticated algorithm with many linear algebraic techniques that are difficult to understand and implement. These techniques were necessitated by parameterizing its state matrix as a **diagonal plus low-rank** (DPLR) matrix, which is necessary to capture HiPPO matrices. A natural question is

36th Conference on Neural Information Processing Systems (NeurIPS 2022).

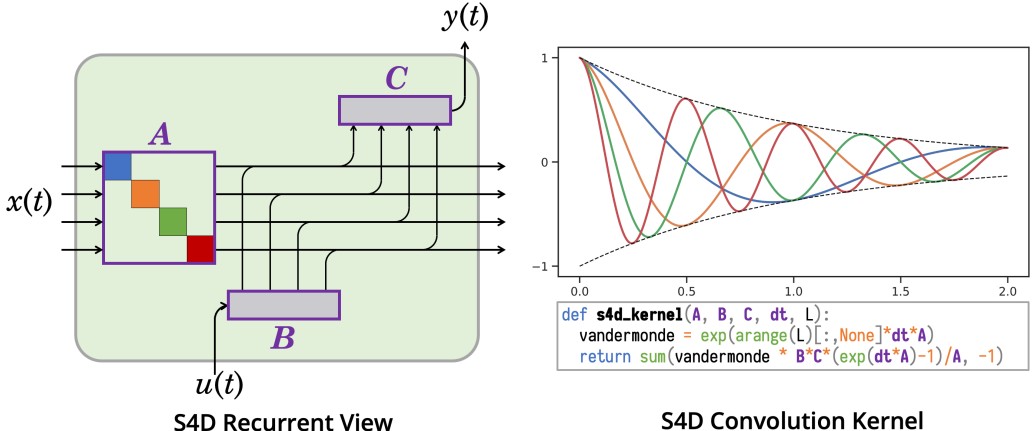

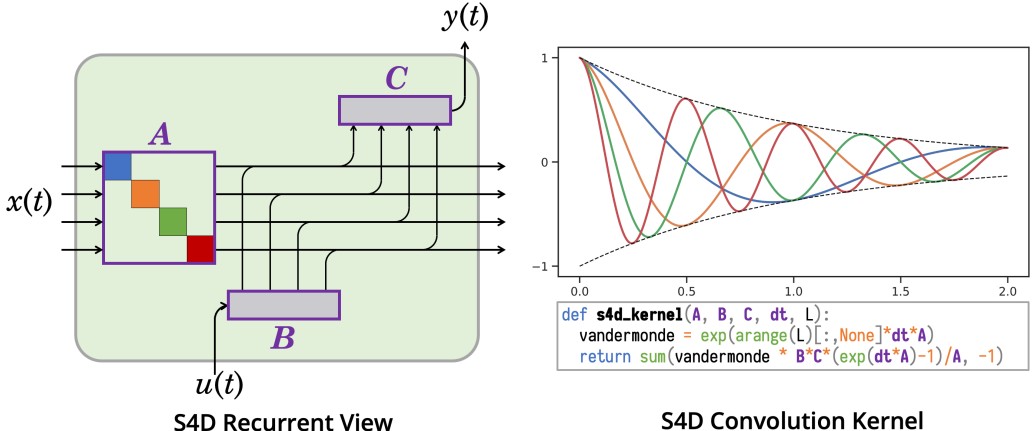

$y(t)$

$C$

$A$

$x(t)$

$B$

$u(t)$

```
def s4d_kernel(A, B, C, dt, L):
    vandermonde = exp(arange(L)[:,None]*dt*A)
    return sum(vandermonde * B*C*(exp(dt*A)-1)/A, -1)
```

S4D Recurrent View    S4D Convolution Kernel

Figure 1: S4D is a diagonal SSM which inherits the strengths of S4 while being much simpler. (*Left*) The diagonal structure allows it to be viewed as a collection of 1-dimensional SSMs. (*Right*) As a convolutional model, S4D has a simple interpretable convolution kernel which can be implemented in two lines of code. Colors denote independent 1-D SSMs; purple denotes trainable parameters.

whether simplifications of this parameterization and algorithm are possible. In particular, removing the low-rank term would result in a **diagonal state space model** (diagonal SSM) that is dramatically simpler to implement and understand.

Although it is known that almost all SSMs have an equivalent diagonal form—and therefore (complex) diagonal SSMs are fully expressive algebraically—they may not represent all SSMs numerically, and finding a good initialization is critical. Gu et al. [9] showed that it is difficult to find a performant diagonal SSM, and that many alternative parameterizations of the state matrix – including by random diagonal matrices – are much less effective empirically, which motivated the necessity of the more complicated HiPPO matrix. However, recently Gupta [11] made the empirical observation that a variant of S4 using a *particular diagonal matrix* is nearly as effective as the original S4 method. This matrix is based on the original HiPPO matrix and is defined by simply chopping off the low-rank term in the DPLR representation.

The discovery of performant diagonal state matrices opens up new possibilities for simplifying deep state space models, and consolidating models such as S4 and DSS to understand and improve them. First, the strongest version of DSS *computes* the SSM with a complex-valued softmax that complicates the algorithm, and is actually less efficient than S4. Additionally, DSS and S4 differ in several auxiliary aspects of *parameterizing* SSMs that can conflate performance effects, making it more difficult to isolate the core effects of diagonal versus DPLR state matrices. Most importantly, DSS relies on *initializing* the state matrix to a particular approximation of S4's HiPPO matrix. While S4's matrix has a mathematical interpretation for addressing long-range dependencies, the efficacy of the diagonal approximation to it remains theoretically unexplained.

In this work, we seek to systematically understand how to train diagonal SSMs. We introduce the **S4D** method, a diagonal SSM which combines the best of S4's *computation* and *parameterization* and DSS's *initialization*, resulting in a method that is extremely simple, theoretically princpled, and empirically effective.

- First, we describe S4D, a simple method outlined by S4 for computing diagonal instead of DPLR matrices, which is based on **Vandermonde matrix multiplication** and is even simpler and more efficient than the DSS. Outside of the core state matrix, we categorize different representations of the other components of SSMs, introducing flexible design choices that capture both S4 and DSS and allow different SSM parameterizations to be systematically compared (Section 3).

- We provide a new mathematical analysis of DSS's initialization, showing that the diagonal approximation of the original HiPPO matrix surprisingly produces the same dynamics as S4 when the state size goes to infinity. We propose even simpler variants of diagonal SSMs using different initializations of the state matrix (Section 4).

- We perform a controlled study of these various design choices across many domains, tasks, and sequence lengths, and additionally compare diagonal (S4D) versus DPLR (S4) variants. Our best S4D methods are competitive with S4 on almost all settings, with near state-of-the-art results on

image, audio, and medical time series benchmarks, and achieving **85%** on the Long Range Arena benchmark (Section 5).

## 2 Background

**Continuous State Spaces Models**    S4 investigated state space models (1) that are parameterized maps on signals $u(t) \mapsto y(t)$. These SSMs are linear time-invariant systems that can be represented either as a linear ODE (equation (1)) or convolution (equation (2)).

$$
\begin{aligned}
x'(t) &= \boldsymbol{A}x(t) + \boldsymbol{B}u(t) \\
y(t) &= \boldsymbol{C}x(t)
\end{aligned}
\qquad (1)
\qquad
\begin{aligned}
K(t) &= \boldsymbol{C}e^{t\boldsymbol{A}}\boldsymbol{B} \\
y(t) &= (K * u)(t)
\end{aligned}
\qquad (2)
$$

Here the parameters are the state matrix $\boldsymbol{A} \in \mathbb{C}^{N \times N}$ and other matrices $\boldsymbol{B} \in \mathbb{C}^{N \times 1}, \boldsymbol{C} \in \mathbb{C}^{1 \times N}$. In the case of diagonal SSMs, $\boldsymbol{A}$ is diagonal and we will overload notation so that $\boldsymbol{A}_n, \boldsymbol{B}_n, \boldsymbol{C}_n$ denotes the entries of the parameters.

An intuitive way to view the convolution kernel (2) is to interpret it as a linear combination (controlled by $\boldsymbol{C}$) of **basis kernels** $K_n(t)$ (controlled by $\boldsymbol{A}, \boldsymbol{B}$)

$$
K(t) = \sum_{n=0}^{N-1} \boldsymbol{C}_n K_n(t) \qquad\qquad K_n(t) := \boldsymbol{e}_n^\top e^{t\boldsymbol{A}} \boldsymbol{B} \qquad (3)
$$

We denote this basis as $K(t) = K_{\boldsymbol{A},\boldsymbol{B}}(t) = e^{t\boldsymbol{A}}\boldsymbol{B}$ if necessary to disambiguate; note that it is a vector of $N$ functions. In the case of diagonal SSMs, each function $K_n(t)$ is just $e^{t\boldsymbol{A}_n}\boldsymbol{B}_n$.

**S4: Structured State Spaces**    As a deep learning model, SSMs have many elegant properties with concrete empirical and computational benefits [8]. For example, the convolutional form (2) can be converted into a temporal recurrence that is substantially faster for autoregressive applications [5].

However, making SSMs effective required overcoming two key challenges: choosing appropriate values for the matrices, and computing the kernel (2) efficiently.

First, Gu et al. [8] showed that naive instantiations of the SSM do not perform well, and instead relied on a particular (real-valued) matrix $\boldsymbol{A}$ called the HiPPO-LegS matrix (4).[1] These matrices were derived so that the basis kernels $K_n(t)$ have closed-form formulas $L_n(e^{-t})$, where $L_n(t)$ are normalized Legendre polynomials. Consequently, the SSM has a mathematical interpretation of decomposing the input signal $u(t)$ onto a set of infinitely-long basis functions that are orthogonal respect to an exponentially-decaying measure, giving it long-range modeling abilities [10].

Second, S4 introduced a particular parameterization that decomposed this $\boldsymbol{A}$ matrix into the sum of a normal and rank-1 matrix (5), which can be unitarily conjugated into a (complex) diagonal plus rank-1 matrix. Leveraging this structured form, they then introduced a sophisticated algorithm for efficiently computing the convolution kernel (2) for state matrices that are **diagonal plus low-rank (DPLR)**.

$$
\boldsymbol{A}_{nk} = -
\begin{cases}
(2n+1)^{\frac{1}{2}}(2k+1)^{\frac{1}{2}} & n > k \\
n+1 & n = k \\
0 & n < k
\end{cases}
\qquad (4)
\qquad
\boldsymbol{A}_{nk}^{(N)} = -
\begin{cases}
(n+\frac{1}{2})^{1/2}(k+\frac{1}{2})^{1/2} & n > k \\
\frac{1}{2} & n = k \\
(n+\frac{1}{2})^{1/2}(k+\frac{1}{2})^{1/2} & n < k
\end{cases}
$$

$$
\boldsymbol{B}_n = (2n+1)^{\frac{1}{2}} \quad \boldsymbol{P}_n = (n+1/2)^{\frac{1}{2}}
\qquad\qquad
\boldsymbol{A} = \boldsymbol{A}^{(N)} - \boldsymbol{P}\boldsymbol{P}^\top, \qquad \boldsymbol{A}^{(D)} := \operatorname{eig}(\boldsymbol{A}^{(N)})
$$

$$
\textbf{(HiPPO-LegS matrix used in S4)}
\qquad\qquad
\textbf{(Normal / diagonal plus low-rank form)}
\qquad (5)
$$

**DSS: Diagonal State Spaces**    S4 was originally motivated by searching for a *diagonal state matrix*, which would be even more structured and result in very simple computation of the SSM. However, the HiPPO-LegS matrix cannot be stably transformed into diagonal form [9, Lemma 3.2], and they were unable to find any diagonal matrices that performed well, resulting in the DPLR formulation.

Gupta [11] made the surprising empirical observation that simply removing the low-rank portion of the DPLR form of the HiPPO-LegS matrix results in a diagonal matrix that performs comparably

---

[1]HiPPO also specifies formulas for $\boldsymbol{B}$, but the state matrix $\boldsymbol{A}$ is more important. There are many other HiPPO instantiations besides LegS, but HiPPO-LegS is the main one that S4 uses and the term "HiPPO matrix" without the suffix refers to this one.

to the original S4 method. More precisely, their initialization is the diagonal matrix $\boldsymbol{A}^{(D)}$, or the diagonalization of $\boldsymbol{A}^{(N)}$ in (5). They termed $\boldsymbol{A}^{(N)}$ the *skew-HiPPO* matrix, which we will also call the *normal-HiPPO* matrix. To be more specific and disambiguate these variants, we may also call $\boldsymbol{A}^{(N)}$ the HiPPO-LegS-N or HiPPO-N matrix and $\boldsymbol{A}^{(D)}$ the HiPPO-LegS-D or HiPPO-D matrix.

In addition to this initialization, they proposed a method for computing a diagonal SSM kernel. Beyond these two core differences, several other aspects of their parameterization differ from S4's.

In Sections 3 and 4, we systematically study the components of DSS: we categorize different ways to parameterize and compute the diagonal state space, and explain the theoretical interpretation of this particular diagonal $\boldsymbol{A}$ matrix.

## 3 Parameterizing Diagonal State Spaces

We describe various choices for the computation and parameterization of diagonal state spaces. Our categorization of these choices leads to simple variants of the core method. Both DSS and our proposed S4D can be described using a combination of these factors (Section 3.4).

### 3.1 Discretization

The true continuous-time SSM can be represented as a continuous convolution $y(t) = (K * u)(t) = \int_0^\infty \boldsymbol{C} e^{s\boldsymbol{A}} \boldsymbol{B} u(t-s)\, ds$.

In discrete time, we view an input sequence $u_0, u_1, \ldots$ as uniformly-spaced samples from an underlying function $u(t)$ and must approximate this integral. Standard methods for doing so that preserve the convolutional structure of the model exist. The first step is to discretize the parameters. Two simple choices that have been used in prior work include

$$(\textbf{Bilinear})\ \ \overline{\boldsymbol{A}} = (\boldsymbol{I} - \Delta/2\boldsymbol{A})^{-1}(\boldsymbol{I} + \Delta/2\boldsymbol{A}) \qquad (\textbf{ZOH})\ \ \overline{\boldsymbol{A}} = \exp(\Delta\boldsymbol{A})$$
$$\overline{\boldsymbol{B}} = (\boldsymbol{I} - \Delta/2\boldsymbol{A})^{-1} \cdot \Delta\boldsymbol{B} \qquad\qquad\qquad \overline{\boldsymbol{B}} = (\Delta\boldsymbol{A})^{-1}(\exp(\Delta \cdot \boldsymbol{A}) - \boldsymbol{I}) \cdot \Delta\boldsymbol{B}.$$

With these methods, the discrete-time SSM output is just

$$y = u * \overline{\boldsymbol{K}} \qquad \text{where } \overline{\boldsymbol{K}} = (\boldsymbol{C}\overline{\boldsymbol{B}}, \boldsymbol{C}\overline{\boldsymbol{A}}\overline{\boldsymbol{B}}, \ldots, \boldsymbol{C}\overline{\boldsymbol{A}}^{L-1}\overline{\boldsymbol{B}}). \tag{6}$$

These integration rules have both been used in prior works (e.g. LMU and DSS use ZOH [26, 11] while S4 and its predecessors use bilinear [6, 8, 9]).

In Section 5, we show that there is little empirical difference between them. However, we note that there is a curious phenomenon where the bilinear transform actually perfectly smooths out the kernel used in DSS to match the S4 kernel (Section 4 Fig. 2d). We additionally note that numerical integration is a rich and well-studied topic and more stable methods of approximating the convolutional integral may exist. For example, it is well-known that simple rules like the Trapezoid rule [18] can dramatically reduce numerical integration error when the function has bounded second derivative.

### 3.2 Convolution Kernel

The main computational difficulty of the original S4 model is computing the convolution kernel $\overline{\boldsymbol{K}}$. This is extremely slow for general state matrices $\boldsymbol{A}$, and S4 introduced a complicated algorithm for DPLR state matrices. When $\boldsymbol{A}$ is diagonal, the computation is nearly trivial. By (6),

$$\overline{\boldsymbol{K}}_\ell = \sum_{n=0}^{N-1} \boldsymbol{C}_n \overline{\boldsymbol{A}}_n^\ell \overline{\boldsymbol{B}}_n \implies \overline{\boldsymbol{K}} = (\overline{\boldsymbol{B}}^\top \circ \boldsymbol{C}) \cdot \mathcal{V}_L(\overline{\boldsymbol{A}}) \qquad \text{where } \mathcal{V}_L(\overline{\boldsymbol{A}})_{n,\ell} = \overline{\boldsymbol{A}}_n^\ell \tag{7}$$

where $\circ$ is Hadamard product, $\cdot$ is matrix multiplication, and $\mathcal{V}$ is known as a **Vandermonde matrix**.

**Time and Space Complexity** The naive way to compute (7) is by materializing the Vandermonde matrix $\mathcal{V}_L(\overline{\boldsymbol{A}})$ and performing a matrix multiplication, which requires $O(NL)$ time and space.

However, Vandermonde matrices are well-studied and theoretically the multiplication can be computed in $\widetilde{O}(N + L)$ operations and $O(N + L)$ space. In fact, Vandermonde matrices are closely related to Cauchy matrices, which are the computational core of S4's DPLR algorithm, and have identical complexity [17].

**Proposition 1.** *The time and space complexity of computing the kernel of diagonal SSMs is equal to that of computing DPLR SSMs.*

We note that on modern parallelizable hardware such as GPUs, a simple fast algorithm is to compute (7) with naive summation (using $O(NL)$ operations), but without materializing the Vandermonde matrix (using $O(N + L)$ space). Just as with S4, this may require implementing a custom kernel in some modern deep learning frameworks such as PyTorch to achieve the space savings.

### 3.3 Parameterization

**Parameterization of $A$.** Note that the kernel $K(t) = Ce^{tA}B$ blows up to $\infty$ as $t \to \infty$ if $A$ has any eigenvalues with positive real part. Goel et al. [5] found that this is a serious constraint that affects the stability of the model, especially when using the SSM autoregressively. They propose to force the real part of $A$ to be negative, also known as the left-half plane condition in classical controls, by parameterizing the real part inside an exponential function $A = -\exp(A_{Re}) + i \cdot A_{Im}$.

We note that instead of $\exp$, any activation function can be used as long as its range is bounded on one side, such as ReLU, softplus, etc. The original DSS does not constrain the real part of $A$, which is sufficient for simple tasks involving fixed-length sequences, but could become unstable in other settings.

**Parameterization of $B, C$.** Another choice in the parameterization is how to represent $B$ and $C$. Note that the computation of the final discrete convolution kernel $\overline{K}$ depends only on the elementwise product $B \circ C$ (equation (7)). Therefore DSS chose to parameterize this product directly, which they call $W$, instead of $B$ and $C$ individually.

However, we observe that this is equivalent to keeping independent $B$ and $C$, and simply freezing $B = 1$ while training $C$. Therefore, just as S4 has separate parameters $A$, $B$, and $C$ and uses a fixed initialization for $A$ and $B$, S4D also proposes separate $A$, $B$, and $C$ and uses fixed initializations for $A$ (discussed in Section 4) and $B$ (set to $1$). Then the difference between S4D and DSS is simply that DSS does not train $B$. In our ablations, we show that training $B$ gives a minor but consistent improvement in performance.

### 3.4 S4D: the Diagonal Version of S4

A key component of our exposition is disentangling the various choices possible in representing and computing state space models. With this categorization, different choices can be mixed and matched to define variants of the core method. Table 1 compares S4, DSS, and S4D, which have a core structure and kernel computation, but have various choices of other aspects of the parameterization.

Table 1: (**Parameterization choices for Structured SSMs**.) Aside from the core structure of $A$ and the computation of its convolution kernel, SSMs have several design choices which are consolidated in S4D.

| Method | Structure | Kernel Computation | Discretization | Constraint $\Re(A)$ | Trainable $B$ | Initialization of $A$ |
|---|---|---|---|---|---|---|
| **S4** | DPLR | Cauchy | Bilinear | exp | Yes | HiPPO |
| **DSS** | diagonal | softmax | ZOH | id (none) | No | HiPPO-D |
| **S4D** | diagonal | Vandermonde | either | exp / ReLU | optional | various |

## 4 Initialization of Diagonal State Matrices

The critical question remains: which diagonal state matrices $A$ are actually effective? We comment on the limitations of diagonal SSMs, and then provide three instantiations of S4D that perform well empirically.

**Expressivity and Limitations of Diagonal SSMs.** We first present a simplified view on the expressivity of diagonal SSMs mentioned by [11]. First, it is well-known that almost all matrices diagonalize over the complex plane. Therefore it is critical to use complex-valued matrices in order to use diagonal SSMs.

**Proposition 2.** *The set $\mathcal{D} \subset \mathbb{C}^{N \times N}$ of diagonalizable matrices is dense in $\mathbb{C}^{N \times N}$, and has full measure (i.e. its complement has measure $0$).*

It is also well known that the state space $(\boldsymbol{A}, \boldsymbol{B}, \boldsymbol{C})$ is exactly equivalent to (i.e. expresses the same map $u \mapsto y$) the state space $(\boldsymbol{V}^{-1}\boldsymbol{A}\boldsymbol{V}, \boldsymbol{V}^{-1}\boldsymbol{B}, \boldsymbol{C}\boldsymbol{V})$, known in the SSM literature as a state space transformation. Therefore Proposition 2 says that *(almost) all SSMs are equivalent to a diagonal SSM*.

However, we emphasize that Proposition 2 is about *expressivity* which does not guarantee strong performance of a trained model after optimization. For example, Gu et al. [9] and Gupta [11] show that parameterizing $\boldsymbol{A}$ as a dense real matrix or diagonal complex matrix, which are both fully expressive classes, performs poorly if randomly initialized.

Second, Proposition 2 does not take into account numerical representations of data, which was the original reason S4 required a low-rank correction term instead of a pure diagonalization [9, Lemma 3.2]. In Section 5.2, we also show that two different initializations with the *same spectrum* (i.e., are equivalent to the same diagonal $\boldsymbol{A}$) can have very different performance.

**S4D-LegS.** The HiPPO-LegS matrix has DPLR representation $\boldsymbol{A}^{(D)} - \boldsymbol{P}\boldsymbol{P}^\top$, and Gupta [11] showed that simply approximating it with $\boldsymbol{A}^{(D)}$ works quite well (5). Our first result is providing a clean mathematical interpretation of this method. Theorem 3 shows a surprising fact that does not hold in general for DPLR matrices (Appendix A.1), and arises out of the special structure of this particular matrix.

**Theorem 3.** *Let $\boldsymbol{A} = \boldsymbol{A}^{(N)} - \boldsymbol{P}\boldsymbol{P}^\top$ and $\boldsymbol{B}$ be the HiPPO-LegS matrices, and $K_{\boldsymbol{A},\boldsymbol{B}}(t)$ be its basis. As the state size $N \to \infty$, the SSM basis $K_{\boldsymbol{A}^{(N)},\boldsymbol{B}/2}(t)$ limits to $K_{\boldsymbol{A},\boldsymbol{B}}(t)$ (Fig. 2).*

Note that $\boldsymbol{A}^{(N)}$ is then *unitarily* equivalent to $\boldsymbol{A}^{(D)}$, which preserves the stability and timescale [10] of the system.

We define **S4D-LegS** to be the S4D method for this choice of diagonal $\boldsymbol{A} = \boldsymbol{A}^{(D)}$. Theorem 3 explains the empirical results in [11] whereby this system performed quite close to S4, but was usually slightly worse. This is because DSS is a variant of S4D-LegS, which by Theorem 3 is a noisy approximation to S4-LegS. Fig. 2 illustrates this result, and also shows a curious phenomenon involving different discretization rules that is open for future work.

**S4D-Inv.** To further simplify S4D-LegS, we analyze the structure of $\boldsymbol{A}^{(D)} = \mathrm{diag}\langle\boldsymbol{A}\rangle$ in more detail. The real part is easy to understand, which follows from the analysis in [9]: $\Re(\boldsymbol{A}) = -\frac{1}{2}\boldsymbol{1}$. Let the imaginary part be sorted, i.e. $\Im(\boldsymbol{A})_n$ is the $n$-th largest (positive) imaginary component. We empirically deduced the following conjecture for the asymptotics of the imaginary part.

**Conjecture 4.** *As $N \to \infty$, $\Im(\boldsymbol{A})_0 \to \frac{1}{\pi}N^2 + c$ where $c \approx 0.5236$ is a constant. For a fixed $N$, the other eigenvalues satisfy an inverse scaling in $n$: $\Im(\boldsymbol{A})_n = \Theta(n^{-1})$.*

Fig. 3 empirically supports this conjecture. Based on Conjecture 4, we propose the initialization S4D-Inv to use the following inverse-law diagonal matrix which closely approximates S4D-LegS.

$$\text{(S4D-Inv)} \quad \boldsymbol{A}_n = -\frac{1}{2} + i\frac{N}{\pi}\left(\frac{N}{2n+1} - 1\right) \quad (8) \quad \text{(S4D-Lin)} \quad \boldsymbol{A}_n = -\frac{1}{2} + i\pi n \quad (9)$$

**S4D-Lin.** While S4D-Inv can be seen as an approximation to the original S4-LegS, we propose an even simpler scaling law for the imaginary parts that can be seen as an approximation of S4-FouT ([10]), where the imaginary parts are simply the Fourier series frequencies (i.e. matches the diagonal part of the DPLR form of S4-FouT). Fig. 1 (*Right*) illustrates the S4D-Lin basis $e^{t\boldsymbol{A}}\boldsymbol{B}$, which are simply damped Fourier basis functions.

## 5 Experiments

Our experimental study shows that S4D has strong performance in a wide variety of domains and tasks, including the well-studied Long Range Arena (LRA) benchmark where the best S4D variant is competitive with S4 on all tasks and significantly outperforms all non-SSM baselines.

We begin with controlled ablations of the various representations of diagonal state space models. Sections 5.1 and 5.2 ablate the proposed methods for parameterizing, computing, and initializing diagonal SSMs from Sections 3 and 4. Section 5.3 show full results of larger models on standard benchmarks,

**Methodology and Datasets.** In order to study the effects of different S4 and S4D variants in a controlled setting, we propose the following protocol. We focus on three datasets covering a varied

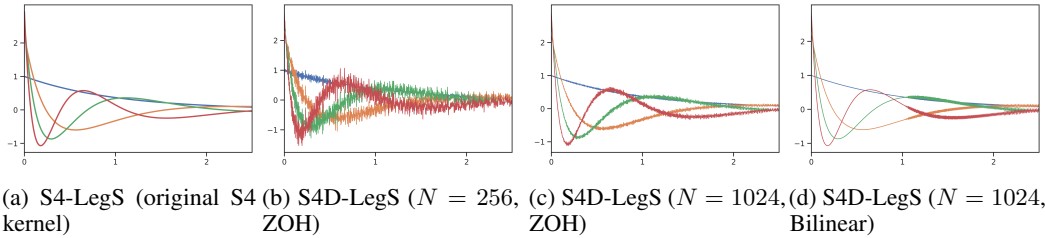

(a) S4-LegS (original S4 kernel)    (b) S4D-LegS ($N = 256$, ZOH)    (c) S4D-LegS ($N = 1024$, ZOH)    (d) S4D-LegS ($N = 1024$, Bilinear)

Figure 2: **(Visualization of Theorem 3).** (a) The particular $(\boldsymbol{A}, \boldsymbol{B})$ matrix chosen in S4 results in smooth basis functions $e^{t\boldsymbol{A}}\boldsymbol{B}$ with a closed form formula in terms of Legendre polynomials. By the HiPPO theory, convolving against these functions has a mathematical interpretation as orthogonalizing against an exponentially-decaying measure. (b, c) By special properties of this state matrix, removing the low-rank term of its NPLR representation produces the same basis functions as $N \to \infty$, explaining the empirical effectiveness of DSS. (c) Curiously, the bilinear transform instead of ZOH smooths out the kernel to exactly match S4-LegS as $N$ grows.

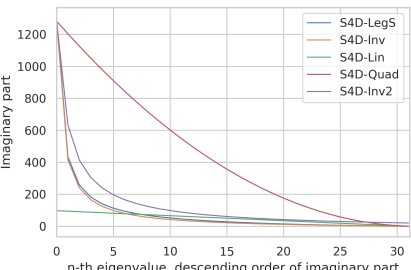

Figure 3: **(S4D eigenvalues.)** All S4D methods have eigenvalues $-\frac{1}{2} + \lambda_n i$. S4D-LegS theoretically approximates dynamics of the original (non-diagonal) S4 (Blue), and has eigenvalues following an inverse law $\lambda_n \propto n^{-1}$ (Orange). The precise law is important: other scaling laws with the same range, including an inverse law with different constant (Purple) and a quadratic law (Red), perform empirically worse (Section 5.2). A very different linear law based on Fourier frequencies also performs well (Green).

range of data modalities (image pixels, biosignal time series, audio waveforms), sequence lengths (1K, 4K, 16K), and tasks (classification and regression with bidirectional and causal models).

- **Sequential CIFAR (sCIFAR).** CIFAR-10 images are flattened into a sequence of length 1024, and a bidirectional sequence model is used to perform 10-way classification.
- **BIDMC Vital Signs.** EKG and PPG signals of length 4000 are used to predict respiratory rate (RR), heart rate (HR), and blood oxygen saturation (SpO2). We focus on SpO2 in this study.
- **Speech Commands (SC).**[2] A 1-second raw audio waveform comprising 16000 samples is used for 35-way spoken word classification. We use an autoregressive (AR) model to vary the setting; this causal setting more closely imitates autoregressive speech generation, where SSMs have shown recent promise [5].

We fix a simple architecture and training protocol that works generically. The architecture has 4 layers and hidden dimension $H = 128$, resulting in $\sim 100K$ parameters. All results are averaged over multiple seeds (full protocol and results including std. reported in Appendix B).

## 5.1 Parameterization, Computation, Discretization

Given the same diagonal SSM matrices $\boldsymbol{A}, \boldsymbol{B}$, there are many variants of how to parameterize the matrices and compute the SSM kernel described in Section 3. We ablate the different choices described in Table 1. Results are in Table 2, and show that:

(i) Computing the model with a softmax instead of Vandermonde product does not make much difference

(ii) Training $\boldsymbol{B}$ is consistently slightly better

(iii) Different discretizations (Section 3.1) do not make a noticeable difference

(iv) Unrestricting the real part of $\boldsymbol{A}$ (Section 3.3) may be slightly better

These ablations show that for a fixed initialization $(\boldsymbol{A}, \boldsymbol{B})$, different aspects of parameterizing SSMs make little difference overall. This justifies the parameterization and algorithm S4D uses (Section 3.4), which preserves the choices of the original S4 model and is simpler than DSS. For the remaining

---

[2]We note that a line of prior work including S4 [14, 19, 9] all used a smaller 10-class subset of SC, so our results on the full dataset are not directly comparable.

| Trainable B | Method | sCIFAR | SC (AR) | BIDMC (SpO2) |
|---|---|---|---|---|
| No | Softmax | 85.04 | 89.80 | 0.1299 |
| No | Vandermonde | 84.78 | 89.62 | 0.1355 |
| Yes | Softmax | 85.37 | 90.06 | 0.1170 |
| Yes | Vandermonde | 85.37 | 90.34 | 0.1274 |

| Discretization | Real part of A | sCIFAR | SC (AR) | BIDMC (SpO2) |
|---|---|---|---|---|
| Bilinear | exp | 85.20 | 89.52 | 0.1193 |
| | ReLU | 85.06 | 90.22 | 0.1172 |
| | - | 85.35 | 90.58 | 0.1102 |
| ZOH | exp | 85.02 | 89.93 | 0.1303 |
| | ReLU | 84.98 | 90.03 | 0.1232 |
| | - | 85.15 | 90.19 | 0.1289 |

Table 2: Ablations of different parameterizations of diagonal SSMs using S4D-Inv. (*Left*) trainability and computation; (*Right*) discretization and parameterization.

of the experiments in Section 5.2 and Section 5.3, we fix the S4D parameterization and algorithm described in Section 3. Note that this computes exactly the same kernel as the original S4 algorithm when the low-rank portion is set to 0, allowing controlled comparisons of the critical state matrix $\boldsymbol{A}$ for the remainder of this section.

## 5.2 S4D Initialization Ablations

The original S4 model proposed a specific formula for the $\boldsymbol{A}$ matrix, and the first diagonal version [11] used a specific matrix based on it. Our new proposed variants S4D-Inv and S4D-Lin also define precise formulas for the initialization of the $\boldsymbol{A}$ matrix (8). This raises the question of whether the initialization of the $\boldsymbol{A}$ still needs to be so precise, despite the large simplifications from the original version. We perform several natural ablations on these initializations, showing that even simple variations of the precise formula can degrade performance.

**Imaginary part scaling factor.** The scaling rules for the imaginary parts of S4D-Inv and S4D-Lin are simple polynomial laws, but how is the constant factor chosen and how important is it? These constants are based on approximations to HiPPO methods (e.g. Conjecture 4). Note that the range of imaginary components for S4D-Inv and S4D-Lin are quite different (Fig. 3); the largest imaginary part is $\frac{N^2}{\pi}$ for S4D-Inv and $\pi N$ for S4D-Lin.

We consider scaling all imaginary parts by a constant factor of $0.01$ or $100.0$ to investigate whether the constant matters. Note that this preserves the overall shape of the basis functions (Fig. 1, dashed lines) and simply changes the frequencies, and it is not obvious that this should degrade performance. However, both changes substantially reduce the performance of S4D in all settings.

**Randomly initialized imaginary part.** Next, we consider choosing the imaginary parts randomly. For S4D-Inv, we keep the real parts equal to $-\frac{1}{2}$ and set each imaginary component to $\boldsymbol{A}_n = -\frac{1}{2} + i\frac{N}{\pi}\left(\frac{N}{2u+1} - 1\right)$ for $u \sim N \cdot \mathcal{U}[0,1]$. Note that when $u$ is equally spaced in $[0,1]$ instead of uniformly random, this exactly recovers S4D-Inv (8), so this is a sensible random approximation to it.

Similarly, we consider a variant of S4D-Lin that replaces the $n$ in (9) with $N \cdot \mathcal{U}[0,1]$.

Table 3a (*Random Imag*) shows that this small change causes minor degradation in performance. We additionally note that the randomly initialized imaginary ablation can be interpreted as follows. Fig. 3 shows the asymptotics of the imaginary parts of SSM matrices, where the imaginary parts of the eigenvalues correspond to y-values corresponding to uniformly spaced nodes on the x-axis. This ablation then replaces the uniform spacing on the x-axis with uniformly random x values.

**Randomly initialized real part.**

We considering initializing the real part of each eigenvalue as $-\mathcal{U}[0,1]$ instead of fixing them to $-\frac{1}{2}$. Table 3a(Left, *Random Real*) shows that this also causes minor but consistent degradation in performance on the ablation datasets. Finally, we also consider randomizing both real and imaginary parts, which degrades performance even further.

**Ablation: Other S4D matrices.**

Other simple variants of initializations show that it is not just the range of the eigenvalues but the actual distribution that is important (Fig. 3). Both S4D-Inv2 and S4D-Quad have real part $-\frac{1}{2}$ and imaginary part satisfying the same maximum value as Conjecture 4. The S4D-Inv2 initialization uses the same formula as S4D-Inv, but replaces a $2n + 1$ in the denominator with $n + 1$. The S4D-Quad initialization uses a polynomial law with power 2 instead of $-1$ (S4D-Inv) or 1 (S4D-Lin).

We include two additional methods here that are not based on the proposed S4D-Inv or S4D-Lin methods. First, S4D-Rand uses a randomly initialized diagonal $\boldsymbol{A}$, and validates that it performs

Table 3: (*Initialization and Trainability ablations*)

| Ablation | sCIFAR | SC (AR) | BIDMC |
|---|---|---|---|
| **S4D-Lin** | **85.12** | **90.66** | **0.128** |
| Scale 0.01 | -7.27 | -1.92 | +0.040 |
| Scale 100 | -7.91 | -4.04 | +0.077 |
| Random Imag | -0.42 | -3.08 | -0.001 |
| Random Real | -0.73 | -0.87 | +0.011 |
| Random Both | -1.28 | -5.88 | +0.007 |
| | | | |
| **S4D-Inv** | **84.79** | **90.27** | **0.114** |
| Scale 0.01 | -5.03 | -0.08 | +0.028 |
| Scale 100 | -7.77 | -52.31 | +0.034 |
| Random Imag | -0.29 | -0.52 | +0.010 |
| Random Real | 0.12 | -2.18 | +0.032 |
| Random Both | -1.55 | -0.55 | +0.024 |
| S4D-Inv2 | -2.62 | -39.84 | +0.005 |
| S4D-Quad | -1.83 | -0.62 | +0.024 |
| S4D-Random | -6.32 | -1.95 | +0.034 |
| S4D-Real | -5.45 | -10.17 | +0.066 |

| | sCIFAR | | SC (AR) | | BIDMC |
|---|---|---|---|---|---|
| **Frozen (A, B)** | Acc (first) | Acc (best) | Acc (first) | Acc (best) | RMSE (best) |
| S4-LegS | 53.63 | 86.19 | 33.87 | 85.33 | 0.1049 |
| S4-LegT | 54.76 | 86.30 | 8.77 | 57.35 | 0.1106 |
| S4-FouT | 55.28 | 86.05 | 9.27 | 69.57 | 0.1072 |
| S4-LegS+FouT | 54.38 | 86.53 | 34.06 | 83.37 | 0.0887 |
| S4D-LegS | 50.87 | 84.81 | 22.76 | 77.18 | 0.0960 |
| S4D-Inv | 53.19 | 84.40 | 18.49 | 76.53 | 0.0995 |
| S4D-Lin | 51.75 | 84.96 | 19.09 | 75.58 | 0.0935 |
| **Trainable (A, B)** | | | | | |
| S4-LegS | 54.23 | 86.29 | 62.19 | 90.68 | 0.1033 |
| S4-LegT | 55.16 | 86.12 | 55.86 | 90.42 | 0.1146 |
| S4-FouT | 55.89 | 85.93 | 60.56 | 90.83 | 0.1136 |
| S4-LegS+FouT | 55.00 | 86.18 | 61.76 | 91.01 | 0.0970 |
| S4D-LegS | 50.41 | 85.64 | 47.54 | 88.47 | 0.1148 |
| S4D-Inv | 53.42 | 84.59 | 45.73 | 89.69 | 0.1132 |
| S4D-Lin | 52.23 | 85.75 | 47.68 | 89.56 | 0.1032 |

(a) Ablations of the initialization of the diagonal $A$ matrix in S4D. Very simple changes that largely preserve the structure of the diagonal eigenvalues all degrade performance.

(b) Results for all S4 and S4D methods on the ablation datasets, when the $A$ and $B$ matrices are either frozen (*Top*) or trained (*Bottom*). Diagonal state matrices are highly competitive with full DPLR versions, achieving strong results on all datasets.

poorly, in line with earlier findings [9, 11]. Second, S4D-Real uses a particular real initialization with $A_n = -(n + 1)$. This is the exact same spectrum as the original S4(-LegS) method, which validates that it is not just the diagonalization that matters, highlighting the limitations of Proposition 2.

## 5.3 Full Comparisons of S4D and S4 Methods

**Trainable $A$, $B$ matrices.**

Table 3b shows the performance of all S4D and S4 variants [10] on the ablations datasets. We observe several interesting phenomena:

(i) Freezing the matrices performs comparably to training them on sCIFAR and BIDMC, but is substantially worse on SC. We hypothesize that this results from $\Delta$ being poorly initialized for SC, so that at initialization models do not have context over the entire sequence, and training $A$ and $B$ helps adjust for this. As further evidence, the *finite window methods* S4-LegT and S4-FouT (defined in [10]) have the most limited context and suffer the most when $A$ is frozen.

(ii) The full DPLR versions are often slightly better than the diagonal version throughout the entire training curve. We report the validation accuracy after 1 epoch of training on sCIFAR and SC to illustrate this phenomenon. Note that this is not a consequence of having more parameters (Appendix B).

**Large models on ablation datasets.**

Finally, we relax the strict requirements on model size and regularization for the ablation datasets, and show the performance of S4 and S4D variants on the test sets with a larger model (architecture and training details in Appendix B) when the model size and regularization is simply increased (Table 4). We note that results for each dataset are better than the original S4 model, which was already state-of-the-art on these datasets [8, 9].

**Long Range Arena.**

We use the same hyperparameter setting for the state-of-the-art S4 model in [10] on the Long Range Arena benchmark for testing long dependencies in sequence models. S4D variants are highly competitive on all datasets except Path-X, and outperform the S4 variants on several of them. On Path-X using this hyperparameter setting with bidirectional models, only S4D-Inv, our simpler approximation to the original S4-LegS model, achieves above random chance, and has an average of 85% on the full LRA suite, more than 30 points better than the original Transformer [24].

Table 4: (**Ablation datasets:** Full results with larger models.) For Speech Commands, we show both an autoregressive model as in the ablations, and an unconstrained bidirectional model.

| MODEL | SCIFAR | SC | | BIDMC | | |
| --- | --- | --- | --- | --- | --- | --- |
| | TEST | AR | BI. | HR | RR | SPO2 |
| S4-LegS | **91.80** (0.43) | **93.60** (0.13) | 96.08 (0.15) | **0.332** (0.013) | 0.247 (0.062) | 0.090 (0.006) |
| S4-FouT | 91.22 (0.25) | 91.78 (0.10) | 95.27 (0.20) | 0.339 (0.020) | 0.301 (0.030) | **0.068** (0.003) |
| S4D-LegS | 89.92 (1.69) | 93.57 (0.09) | 95.83 (0.14) | 0.367 (0.001) | 0.248 (0.036) | 0.102 (0.001) |
| S4D-Inv | 90.69 (0.06) | 93.40 (0.67) | 96.18 (0.27) | 0.373 (0.024) | 0.254 (0.022) | 0.110 (0.001) |
| S4D-Lin | 90.42 (0.03) | 93.37 (0.05) | **96.25** (0.03) | 0.379 (0.006) | **0.226** (0.008) | 0.114 (0.003) |

Table 5: (**Long Range Arena**) Accuracy on full suite of LRA tasks. Hyperparameters in Appendix B.

| MODEL | LISTOPS | TEXT | RETRIEVAL | IMAGE | PATHFINDER | PATH-X | AVG |
| --- | --- | --- | --- | --- | --- | --- | --- |
| S4-LegS | 59.60 (0.07) | 86.82 (0.13) | 90.90 (0.15) | 88.65 (0.23) | 94.20 (0.25) | **96.35** | **86.09** |
| S4-FouT | 57.88 (1.90) | 86.34 (0.31) | 89.66 (0.88) | **89.07** (0.19) | **94.46** (0.24) | ✗ | 77.90 |
| S4D-LegS | 60.47 (0.34) | 86.18 (0.43) | 89.46 (0.14) | 88.19 (0.26) | 93.06 (1.24) | 91.95 | 84.89 |
| S4D-Inv | 60.18 (0.35) | **87.34** (0.20) | **91.09** (0.01) | 87.83 (0.37) | 93.78 (0.25) | 92.80 | 85.50 |
| S4D-Lin | **60.52** (0.51) | 86.97 (0.23) | 90.96 (0.09) | 87.93 (0.34) | 93.96 (0.60) | ✗ | 78.39 |
| S4 (original) | 58.35 | 76.02 | 87.09 | 87.26 | 86.05 | 88.10 | 80.48 |
| Transformer | 36.37 | 64.27 | 57.46 | 42.44 | 71.40 | ✗ | 53.66 |

# 6 Conclusion

State space models based on S4 are a promising family of models for modeling many types of sequential data, with particular strengths for continuous signals and long-range interactions. These models are a large departure from conventional sequence models such as RNNs, CNNs, and Transformers, with many new ideas and moving parts. This work provides a more in-depth exposition for all aspects of working with S4-style models, from their core structures and kernel computation algorithms, to miscellaneous choices in their parameterizations, to new theory and methods for their initialization. We systematically analyzed and ablated each of these components, and provide recommendations for building a state space model that is as simple as possible, while as theoretically principled and empirically effective as S4. We believe that S4D can be a strong generic sequence model for a variety of domains, that opens new directions for state space models theoretically, and is much more practical to understand and implement for practitioners.

**Acknowledgments**

We gratefully acknowledge the support of DARPA under Nos. FA86501827865 (SDH) and FA86501827882 (ASED); NIH under No. U54EB020405 (Mobilize), NSF under Nos. CCF1763315 (Beyond Sparsity), CCF1563078 (Volume to Velocity), and 1937301 (RTML); ONR under No. N000141712266 (Unifying Weak Supervision); the Moore Foundation, NXP, Xilinx, LETI-CEA, Intel, IBM, Microsoft, NEC, Toshiba, TSMC, ARM, Hitachi, BASF, Accenture, Ericsson, Qualcomm, Analog Devices, the Okawa Foundation, American Family Insurance, Google Cloud, Swiss Re, Brown Institute for Media Innovation, Department of Defense (DoD) through the National Defense Science and Engineering Graduate Fellowship (NDSEG) Program, Fannie and John Hertz Foundation, National Science Foundation Graduate Research Fellowship Program, Texas Instruments, and members of the Stanford DAWN project: Teradata, Facebook, Google, Ant Financial, NEC, VMWare, and Infosys. The U.S. Government is authorized to reproduce and distribute reprints for Governmental purposes notwithstanding any copyright notation thereon. Any opinions, findings, and conclusions or recommendations expressed in this material are those of the authors and do not necessarily reflect the views, policies, or endorsements, either expressed or implied, of DARPA, NIH, ONR, or the U.S. Government.

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
