# OpenReview forum: "On the Parameterization and Initialization of Diagonal State Space Models"
_NeurIPS.cc/2022/Conference — NeurIPS 2022 Accept_

### Official Review · Reviewer_s9cB · 2022-06-27

**Rating:** 6
**Confidence:** 2
**Soundness:** 3 good
**Presentation:** 2 fair
**Contribution:** 2 fair

**Summary:**

Detailed analyses of some variants of the so-called S4 model are presented. The focus is on the use of a diagonal state-transition matrix instead of a diagonal plus low-rank matrix, time discretization scheme, spectral radius constraint of $A$, parametrization of $B$ and $C$, normalization of convolution kernels corresponding to SSM, and initialization scheme. The authors provide empirical ablation studies in these regards. They also provide the rationale for why the diagonal approximation of a so-called HiPPO-LegS matrix works well.

**Questions:**

**1.**
In Figures 1 and 2, please specify what is drawn there not only in the caption but also by using axis labels and legends. The current way of presentation is not so kind.

**2.**
How the proposed models, which are sequence-to-sequence models, were applied to the sequence-to-label problems is not clearly stated. Maybe it is just the same as the previous works, but it would be nice to briefly mention the model architecture somewhere in the main text of the paper.

**3.**
The proof of Theorem 2. I found no obvious errors in the step-by-step reasoning. However, the intermediate result of $u(t) = \mathbf{B}^T x(t)$ is quite counterintuitive. I understand this happens only for $N \to \infty$ with the HiPPO-LegS $\mathbf{B}$ matrix, but I wonder if there is any more intuitive interpretation of why such a relation is the case here.

**Limitations:**

I don't have particular comments in this regard.

**Strengths And Weaknesses:**

Overall I did not find serious shortcomings. The analyses would be of interest to a limited community.

### Strengths

- The experiments are intensive and well organized.
- While I could not check all the correctness of the proofs, the discussion around Theorem 2 is interesting.

### Weaknesses

- The target audience is limited.
- The evaluation is done only in terms of the final performance of classification or regression tasks. Some qualitative and direct analyses of the models' outputs, if possible (and perhaps on much simpler input data), would be interesting.

---

> ### Author Response · Authors · 2022-08-02
> **Response to Reviewer s9cB**
>
> We thank the reviewer for their feedback and suggestions, and appreciate the assessment that the theoretical contribution (Theorem 2) is interesting and the experiments are thorough. The reviewer's main concern is that the significance of this work is to a limited target audience. However, we believe that this work can be potentially quite impactful by exposing a larger portion of the machine learning community to the core ideas of deep state space models. These models based off S4 have achieved very strong results on several benchmarks and have garnered interest in the community, but thus far have been difficult to understand for many practitioners. We hope that the extremely simple models introduced by S4D can open the door to wider adoption of S4, and eventually even simpler and more performant sequence models.
>
> > Some qualitative and direct analyses of the models' outputs, if possible (and perhaps on much simpler input data), would be interesting.
>
> As we also noted to Reviewer E3uH, such analyses is outside the scope and space limitations of this submission, but we agree that understanding the model's internal representations is a valuable direction for future work.
>
> > In Figures 1 and 2, please specify what is drawn there not only in the caption but also by using axis labels and legends. The current way of presentation is not so kind.
>
> It is difficult to include in the legend, but the revised paper has a more detailed description of these figures. In short, the lines represent basis functions defined in equation (3), or the elements of the vector $e^{t\mathbf{A}}\mathbf{B}$ (which are functions)
>
> > How the proposed models, which are sequence-to-sequence models, were applied to the sequence-to-label problems is not clearly stated. Maybe it is just the same as the previous works
>
> The revised version describes the architecture in more detail at the end of Section 3.4; it is indeed the same as previous works. Following prior work (including standard models such as CNNs), the sequence-to-label setting is handled by simply average pooling over the sequence dimension.
>
> > The proof of Theorem 2… is quite counterintuitive
>
> Our best intuition at present is the following, based on HiPPO theory. Since $x(t)$ contains enough information for an (approximate) reconstruction of the entire history of $u$, the current value of $u(t)$ can also be (approximately) extracted by a linear projection of $x(t)$. This projection $\mathbf{B}^\top x(t)$ will be a noisy approximation of $u(t)$ that gets more precise as $N \to \infty$.

---

### Official Review · Reviewer_zJVF · 2022-07-07

**Rating:** 7
**Confidence:** 3
**Soundness:** 4 excellent
**Presentation:** 3 good
**Contribution:** 3 good

**Summary:**

Paper simplifies diagonal state space models by i) analysing Eigen-values of Hippo matrices and proposing appropriate initialisation schemes, ii) simplifying the parameterisation of the real parts (to be negative) of the Eigen-values (for stability) and iii) introducing a simple normalisation scheme that ensures variance preservation of the resulting SSM. Experiments are conducted to verify all these simplifications and the authors show that they do not lead to inferior performance. A very important take away message from the paper is that diagonalization alone doesn’t make a performant SSM but it needs proper parameterisation/initialization. Overall the paper is a great extension to the growing literature on SSMs for Deep Learning.

**Questions:**

* “Second, Proposition 1 does not take into account numerical representations of data, which was  the original reason S4 required a low-rank correction term instead of a pure diagonalization.“   —> This is quite vague, pleas elaborate or strike.
* How do you go from conjecture 3 to S4D-Inv?
* Table 1: What S4D variant is used?
* Curious to see what would happen if only A were left untrained.
* S4D-Trap introduced but never ablated?
* More information on how robust S4D is to choice of optimization hparams is would be interesting.
* Simplicity is really nice but it is not clear to me from the paper why S4D should be used over S4 for any practical reasons? Do you see any speed ups? Results seem to still lag behind S4.

**Limitations:**

The authors haven't addressed the limitations.

**Strengths And Weaknesses:**

## Strengths
* good and mostly clear presentation
* solid mathematical foundation for simplifications
* convincing empirical results

## Weaknesses
* No dicusssion/limitations
* Mostly small scale, toyish tasks evaluated
* The results don't show any advantages of S4D over S4

---

> ### Author Response · Authors · 2022-08-02
> **Response to Reviewer zJVF**
>
> We are glad that the reviewer found the paper clear, and valuable for the growing literature on deep state space models. The reviewer's main concerns were about scope of the experiments and the comparison between S4D and S4.
>
> > No dicusssion/limitations
>
> A conclusion section has been added to the revised draft; we encourage the reviewer to read this short paragraph, which also reiterates the strengths of S4D.
>
> > Mostly small scale, toyish tasks evaluated
>
> As the original S4 model was already shown to have strong performance on a wide variety of tasks, the scope of this work was purposefully focused on simplifying S4 and providing an in-depth exposition of the main components of deep SSMs. We hope this allows future researchers to bring the strengths of SSMs to new applications.
>
> Additionally, the results in Section 5.3 use much larger models that show how the ablations transfer to more realistic settings.
>
> > The results don't show any advantages of S4D over S4
>
> Since S4D is dramatically simpler than S4, we believe that it is not a weakness that it is slightly worse in some settings. The primary contribution of this model is nearly matching the state-of-the-art performance of S4 with a much simpler kernel. We believe that this is an important contribution to the community that will encourage the adoption of state space models.
>
> ### Detailed Questions
>
> > “Second, Proposition 1 does not take into account numerical representations of data, which was the original reason S4 required a low-rank correction term instead of a pure diagonalization.“
>
> This sentence means that some matrices cannot be diagonalized using floating point numbers. The original motivation of S4 was that the HiPPO matrix can be diagonalized only with exponentially large numbers; thus it is effectively not diagonalizable and required the DPLR representation (S4 paper, Section 3.1).
>
> > How do you go from conjecture 3 to S4D-Inv?
>
> This scaling rule was found through exploration - effectively, a mathematical trial-and-error process trying to match a curve to the empirical LegS eigenvalue distribution. Figure 3 shows that it matches the curve quite closely.
>
> > Table 1: What S4D variant is used?
>
> This table used S4D-Inv; it has been added to the caption.
>
> > Curious to see what would happen if only A were left untrained.
>
> We never experimented with freezing A and training B, since our SSM theory always couples $(A, B)$.
>
> > S4D-Trap introduced but never ablated?
>
> The revised draft removes the introduction of S4D-Trap. The third row of the table instead contrasts DSS/S4D with S4, which provides a clearer categorization of the differences
>
> > More information on how robust S4D is to choice of optimization hparams is would be interesting.
>
> The hyperparameters were largely transferred directly from baseline S4 models. As remarked in Appendix B, all S4(D) models were fairly robust to optimization hyperparameters; the same architecture and set of hparams worked quite well for all tasks, with minor deviations for some tasks explained in the Appendix.
>
> > Simplicity is really nice but it is not clear to me from the paper why S4D should be used over S4 for any practical reasons? Do you see any speed ups? Results seem to still lag behind S4.
>
> While simplicity may not be important for a pure applied practitioner, we believe that **simplicity is very important for the research community**. A primary goal of this work is to make the ideas behind deep SSMs understandable to more machine learning researchers and foster this research direction.
>
> Additionally, there actually are several concrete benefits that are mentioned in the revised draft. For example, the parameterization of S4 is tied to a specific sequence length $L$, while S4D can be used more easily on arbitrary sequence lengths. An example future direction mentioned by Reviewer E3uH is to use alternative computation methods such as parallel scans, which are much faster for diagonal SSMs.

---

### Official Review · Reviewer_MC9C · 2022-07-11

**Rating:** 5
**Confidence:** 4
**Soundness:** 3 good
**Presentation:** 2 fair
**Contribution:** 2 fair

**Summary:**

The authors examine a state-space modeling approach that has gained popular in recent literature — the main idea of that line of research concerned with the expressive capability, and ability to capture long-term memory dependence through the use of linear SSMs defined in continuous time.  The authors note that practical use of HiPPo, the structured state-space architecture (S4) and diagonal state-space architectures (DSS) may present complications due to various numerical instabilities associated with bad initializations or parameterizations. To be more specific, they systematically analyze the diagonal approximation to S4 — they prescribe different initializations that perform better numerically for which they also back up theoretically.  Through an ablation study, they are also able to demonstrate the practical utility of their analysis and suggestions.


**Questions:**

If my statement about the dimensionality of the variables specified was wrong, hopefully you could clear that up.  In, Eq. (5) A^D as defined, is always just the matrix of 1/2 on the diagonal, is that correct?

I don’t think the DSS acronym was specified in text (is it diagonal state-space ?).  Is bar{C} as defined in Eq. (6) just equal to the original C row vector?

In Eq. (7) should the ordering of B and C be swapped? Stemming from my confusion earlier, I thought that B was a column vector and C a row vector.  Maybe subscripts here would help i.e. y_l = (u * K)_l or something more economical that clears up confusion.  A reference for the equality using the Vandermonde matrix would probably help as well in my opinion.

On page 5, section 4 below prop. 1 you call the randomly initialized matrices fully expressive — in what sense?


**Limitations:**

In my opinion, the contributions of the paper are somewhat incremental;  that’s not to say the authors contributions are not novel — but a great deal of writing space is used to describe the previous research on HiPPo.  I think more of the authors contributions could have been thrown into the spotlight, rather than to rehash the cited literature.  Furthermore, since a large part of the paper is about how to select good, in some sense, dynamics matrices — how come there is no discussion on parameter training? I would think for a practitioner, it may be easier to train the parameters of the matrices in question, rather than doing a large amount of model selection.

**Strengths And Weaknesses:**

I think the paper addresses some general issues that would be good for someone looking to employ the previously introduced HiPPo matrix into their modeling schemes.  For example, the authors address numerical concerns that are often brushed over in many papers — even though not addressing them can be catastrophic i.e. parameterization of the dynamics such that the matrix is stable.  The memory saving tip in 3.3 for storing B and C is useful as well.  In section 3.4 I like the initialization prescribed, and while its a nice concept I think the presentation could just be slightly tidied up — a comparison to balancing would be useful too, see [E. E. Osborne, “On pre-conditioning of matrices”].  I thought some of the ablation studies were also very useful; since the point of the paper is to prescribe practical guidelines to follow its nice the authors swept through different variables such as the discretization method, normalization, and parameterizations.

One of the first difficulties I encountered was with notation.  If inline with the papers cited regarding HiPPo, then it would have been useful to specify that for example, C is a row vector, B is a column vector, and y(t) and u(t) are one-dimensional observations and control inputs respectively.   The lack of conclusion is also somewhat concerning, I think a succinct summary of the experimental and theoretical results would have been a good way to end the paper.  Another big weakness I think lays in the total contributions of the authors; a lot of rehashing of previous results is done, but the authors in my opinion do not devote enough space in the main text to outline their current contributions.

---

> ### Author Response · Authors · 2022-08-02
> **Response to Questions**
>
> ### Detailed Questions
>
> > If my statement about the dimensionality of the variables specified was wrong, hopefully you could clear that up. In, Eq. (5) A^D as defined, is always just the matrix of 1/2 on the diagonal, is that correct?
>
> $A^{(D)}$ is defined as the diagonalization or eigenvalues of $A^{(N)}$. The notation has been cleared up in the revision (Section 2, paragraph "DSS")
>
> > I don’t think the DSS acronym was specified in text (is it diagonal state-space ?).
>
> DSS was defined in Section 2 in the original submission via the paragraph header "DSS: Diagonal State Spaces"
>
> > Is bar{C} as defined in Eq. (6) just equal to the original C row vector?
>
> Yes. This notation was borrowed from the original S4 paper, but the revision changes $\bar{C}$ to just $C$
>
> > In Eq. (7) should the ordering of B and C be swapped? Stemming from my confusion earlier, I thought that B was a column vector and C a row vector. Maybe subscripts here would help i.e. y_l = (u * K)_l or something more economical that clears up confusion. A reference for the equality using the Vandermonde matrix would probably help as well in my opinion.
>
> The description of the Vandermonde matrix kernel has been substantially improved in the revision. You are correct about B being a column vector and C being a row vector. Moving the code listing to the main body should also clarify these details.
>
> > ​​On page 5, section 4 below prop. 1 you call the randomly initialized matrices fully expressive — in what sense?
>
> "Fully expressive" refers to "dense real matrix or diagonal complex matrix". These classes of matrices are fully expressive in the sense that they are equivalent to (nearly all) real square matrices by conjugation (Proposition 1 in submission, or Proposition 2 in revision).

---

> ### Author Response · Authors · 2022-08-02
> **Response to Reviewer MC9C**
>
> We sincerely appreciate the reviewer's detailed review and constructive feedback. The reviewer asked many detailed questions and raised valid concerns about the presentation and contributions of this work. We respond to these points below, and highlight where the concerns were addressed in the original submission. Additionally, we point to where these aspects have been substantially improved in the revised draft.
>
>
> > Another big weakness I think lays in the total contributions of the authors; a lot of rehashing of previous results is done, but the authors in my opinion do not devote enough space in the main text to outline their current contributions.
>
> > In my opinion, the contributions of the paper are somewhat incremental; that’s not to say the authors contributions are not novel — but a great deal of writing space is used to describe the previous research on HiPPo. I think more of the authors contributions could have been thrown into the spotlight, rather than to rehash the cited literature.
>
> While the reviewer's primary concern is about contributions, we believe that the paper contains several novel contributions that are important for this line of work on state space models. These include (1) a categorization of important SSM components (2) a much simpler and more efficient diagonal kernel based on Vandermonde multiplication (3) new theory explaining the effectiveness of diagonal SSM initializations (4) careful empirical studies to validate these contributions.
>
> Additionally, we emphasize that this work is purposefully expository. For example, several aspects of the parameterization of SSMs such as the discretization were elided over in the original presentation of S4, and are foreign to much of the machine learning community. Although recent deep SSMs based on S4 have garnered substantial interest, they introduce many new concepts that have been a barrier to their adoption. We believe that carefully explaining and ablating these concepts is an important contribution to the community that can increase the visibility of SSM models.
>
> Overall, while the in-depth explanations in this work may be construed as re-hashing previous research on HiPPO and SSMs, we argue that they actually provide novel and valuable exposition to most of the machine learning community, and our categorization of prior concepts is an important contribution in itself.
>
>
> > Furthermore, since a large part of the paper is about how to select good, in some sense, dynamics matrices — how come there is no discussion on parameter training? I would think for a practitioner, it may be easier to train the parameters of the matrices in question, rather than doing a large amount of model selection.
>
> The SSM parameters are in fact trained.
> - All of the empirical ablations train the matrices by default, which show that there is still a substantial difference between initializations even after training, and affirms the importance of the initialization theory proposed in this work.
> - The discussion about initialization (below Proposition 1 in the original, or Proposition 2 in the revision) explains in more detail why initialization is important, even when the matrices are trained.
> - Finally, one of the ablations specifically compares performance when the $(A, B)$ matrices are trained vs frozen (Table 2(b) in the original submission; Table 3(b) in the revision).
>
> The revised draft is also more clear about trainability of the matrices, and has a dedicated discussion in Section 3 to the trainability of B (revision, Table 1).
>
> > One of the first difficulties I encountered was with notation. If inline with the papers cited regarding HiPPo, then it would have been useful to specify that for example, C is a row vector, B is a column vector, and y(t) and u(t) are one-dimensional observations and control inputs respectively.
>
> This is correct. The dimensionalities have been clarified in the revised draft, right under equation (2)
>
> > The lack of conclusion is also somewhat concerning, I think a succinct summary of the experimental and theoretical results would have been a good way to end the paper.
>
> A conclusion section has been added to the revised draft. **We highly encourage the reviewer to read this short paragraph**, which also reiterates the strengths of S4D and contributions of this work.

---

> > ### Comment · Reviewer_MC9C · 2022-08-06
> > **Response**
> >
> > Thank you for the detailed reply.  I appreciate the fixing of the notation, the added details, and additions to serve to make the exposition clearer.  The addition of the code I think was very useful, and so were the changes to Table 1 -- I think the new formatting is more informative.  For theorem 6 in the supplementary material is there a proof anywhere? I don't think its extremely obvious that there should be no proof/reference.  This is tangential, but have the authors considered experiments where observations are read out from some highly non-linear low-dimensional system? In this case, I really expect the required dimensionality of the linear dynamical system necessary to perform accurate inference/prediction will be highly detrimental.  This is definitely one of my biggest concerns with the paper -- these limitations I do not think are clearly addressed.  However, I am raising my score as the authors have addressed many of my concerns.

---

> > > ### Author Response · Authors · 2022-08-09
> > > **Response**
> > >
> > > We appreciate the response and further suggestions.
> > >
> > > > For theorem 6 in the supplementary material is there a proof anywhere? I don't think its extremely obvious that there should be no proof/reference.
> > >
> > > The proof is in the reference "How to Train Your HiPPO" cited right before Theorem 6. A copy of the manuscript has been provided in the latest supplementary material.
> > >
> > > Additional, we have provided the notebook `ssm_kernels.ipynb` in the supplementary. The first 4 cells there verify the claim in Theorem 6 through plotting the SSM kernel $e^{t \mathbf{A}} \mathbf{B}$ as well as the closed form formula $L_n(e^{-t})e^{-t}$ (for Legendre polynomials $L_n$)
> > >
> > > > This is tangential, but have the authors considered experiments where observations are read out from some highly non-linear low-dimensional system? In this case, I really expect the required dimensionality of the linear dynamical system necessary to perform accurate inference/prediction will be highly detrimental. This is definitely one of my biggest concerns with the paper -- these limitations I do not think are clearly addressed.
> > >
> > > The reviewer's new concern is about the efficacy of S4D on non-linear low-dimensional systems. We make three responses to this concern:
> > > 1. One reason why S4 can capture non-linear dynamics has been addressed in its predecessor works [1, Lemma 3.2]: although each layer is linear, the entire deep neural network is highly non-linear through activations in the depth direction, which can locally capture temporal non-linearities.
> > > 2. A primary empirical goal of this work is **showing that S4D is competitive with S4**, in particular on a multitude of tasks on varied domains, sequence lengths, and characteristics (e.g. images, audio, time series, LRA) where S4 is a state-of-the-art model. On the other hand, since it would be infeasible to evaluate S4D in every possible setting, and S4 was never tested on low-dimensional non-linear dynamical systems, we believe it is somewhat outside the scope of this work to focus on these tasks.
> > > 3. With that said, the reviewer's suggestion is an interesting task different than the ones we have considered so far, and we have run preliminary experiments. We used a harder version of the "Mackey-Glass Prediction" task from [2], (a predecessor of HiPPO/S4), described as "a time-series prediction task that tests the ability of a network to model chaotic dynamical systems" [2]. Data is generated by simulating the [Mackey-Glass equation](https://en.wikipedia.org/wiki/Mackey%E2%80%93Glass_equations) for $T=5000$ units of time with a delay of $\tau=150$ and a step size of $\Delta=0.1$. The task is to predict $60$ units into the future, measured by MSE. The following are results for some models, showing that S4D-Lin is close to S4 and both are better than standard baselines.
> > >
> > > | Model | Parameters | Val MSE @ 100 steps | Val MSE @ 1000 steps |
> > > |--------|----------------|-------------------------|---------------------------|
> > > | LSTM | 133313  | 0.109 | 0.00099 |
> > > | GRU | 100033 | 0.105 | 0.00252 |
> > > | S4 | 51777 | 0.023 | 0.00027 |
> > > | S4D-Lin | 51265 | 0.031 | 0.00044 |
> > >
> > >
> > > [1] Gu et al. "Combining Recurrent, Convolutional, and Continuous-Time Models with the Linear State Space Layer." NeurIPS 2021
> > > [2] Voelker et al. "Legendre Memory Units: Continuous-Time Representation in Recurrent Neural Networks". NeurIPS 2019

---

### Official Review · Reviewer_E3uH · 2022-07-21

**Rating:** 8
**Confidence:** 3
**Soundness:** 4 excellent
**Presentation:** 3 good
**Contribution:** 4 excellent

**Summary:**

The authors set out to understand why some diagonal variants of the S4 state-space model work well, then introduce S4D as a specific approach building off of DSS. Their main mathematical result shows that a purely diagonal update reproduces the dynamics of S4’s diagonal plus low rank update in the limit of large latent size.

Based on their theoretical insights, they improve on existing diagonal state-space models (DSS) by carefully choosing initialization, discretization, and related parameters in the S4D family of models. The authors also report detailed ablations to confirm the theoretical conclusions, and demonstrate S4-equivalent quality on a variety of benchmarks.


**Questions:**

Do you have a guess or narrative for what goes wrong when a parameterization is expressive but doesn’t optimize well? (“[W]e emphasize that Proposition 1 is about expressivity which does not guarantee strong performance of a trained model after optimization. For example, Gu et al. [7] and Gupta [8] show that randomly initialized dense real matrices or diagonal complex matrices, which are both fully expressive, perform much worse than S4.”)

The ability to fully diagonalize the SSM update creates a strongly privileged basis; have the authors explored interpretability of latent states in this basis?


**Limitations:**

If there are significant limitations of diagonal SSMs, including S4D, relative to the original S4, it would certainly be worth mentioning them as negative results. But so far it appears not to be the case.

**Strengths And Weaknesses:**

The paper is clear despite packing information very densely and including both theoretical and empirical results.

The ablations are strong and careful, and presented concisely and clearly (I haven’t seen this particular graphical presentation of ablations before, and I think it works well.)

Overall, I think it was unclear before this paper why diagonal SSMs worked but related methods like linear RNNs didn’t; this paper begins to answer that question.

The approach presented in this paper is to compute the kernel using a Vandermonde matrix, but the authors don't make the connection between this approach and the parallel scan implementation of the linear recurrence. The parallel scan approach has some advantages over explicitly computing the kernel, including support for unbounded sequence length (important for online tasks like reinforcement learning) and variable/dynamic sampling rate.

The abstract mentions “three lines of code”; these should be included in the main paper.

---

> ### Author Response · Authors · 2022-08-02
> **Response to Reviewer E3uH**
>
> We are glad that the reviewer found the paper clear, with dense information and careful empirical results. We agree with the reviewer that the connection between diagonal SSMs and linear RNNs is important and hope that the community continues exploring this promising family of models.
>
> We respond to some of the reviewer's detailed points below.
>
> > The parallel scan approach has some advantages over explicitly computing the kernel, including support for unbounded sequence length (important for online tasks like reinforcement learning) and variable/dynamic sampling rate.
>
> We believe that the parallel scan method for RNNs is a very promising approach, and are particularly excited about the support for variable sampling rates, which is one aspect of traditional RNNs such as LSTMs that S4 currently lacks. Additionally, parallel scan is much slower for the original non-diagonal parameterization of S4; we hope that the introduction of performant diagonal SSMs such as S4D can make this approach more realizable as a future direction.
>
> > The abstract mentions “three lines of code”; these should be included in the main paper.
>
> The "three lines of code" is included in the Appendix (Listing 1) of the submission, and is moved to the main body in the revision in Section 3.4. Note that these three lines concern the core kernel construction; a few additional lines are needed for the initialization and convolution.
>
> > Do you have a guess or narrative for what goes wrong when a parameterization is expressive but doesn’t optimize well?
>
> Optimization is still a poorly understood topic in deep learning theory. Our intuition is that generally, good SSM initializations for $(A, B)$ should work fairly well even *without training*, as shown in Table 2(b) (Table 3(b) in the revision). As described in equation (3), the $C$ parameter takes a linear combination of basis functions; poorly initialized $(A, B)$ matrices will create less well-behaved basis functions, whereas the proposed initializations create nice interpretable basis functions (Fig. 1).
>
> > The ability to fully diagonalize the SSM update creates a strongly privileged basis; have the authors explored interpretability of latent states in this basis?
>
> Thus far we have primarily focused on developing models with strong downstream performance. We agree that better understanding the internal representations of these models is an important future direction for this line of work.

---

### Author Response · Authors · 2022-08-02
**Response to all Reviewers**

We thank all reviewers for their time and thoughtful feedback. All reviewers agree that the paper introduces novel theoretical contributions and extensive empirical ablations for the growing line of work on state space models. We sincerely appreciate the detailed questions and suggestions from all reviewers, which have substantially improved the submission, and have been responded to in individual comments.

We have uploaded a revised version of this work that takes into account the reviewer feedback and incorporates numerous other changes such as
- Clearer description of the contributions in the Introduction, with a new splash figure
- Improved categorization of the SSM parameterizations in Section 3
- More detailed description of the Vandermonde kernel, with comparisons to prior work
- Addition of a proper Conclusion section

Due to space constraints, this revision has been uploaded in **anonymized pre-print form in the supplementary material**, but can be readily cut down to NeurIPS format provided additional space in the potential camera ready format.

---

### Meta-Review · Area_Chair_S37Y · 2022-08-24

**Recommendation:** Accept
**Confidence:** Certain

**Metareview:**

All reviewers agree that the paper proposes an interesting approach to examine diagonal state space models. Although some reviewers have some technical concerns at their first reviews, basically those have been resolved by the authors' responses. Thus, although there are some points that should be modified from the current form, I think we can expect the authors modify the paper in the camera-ready by reflecting the discussion. Based on these, I recommend acceptance for this paper.

**Award:**

No

---

### Decision · Program_Chairs · 2022-09-14

Accept